Extended Abstract Track

# On Rotational Symmetry
# in the Loss landscape of Self-Supervised Learning

**Liu Ziyin**[1,2,3]
**Ekdeep Singh Lubana**[2,3,4]
**Masahito Ueda**[1,5,6]
**Hidenori Tanaka**[2,3]

[1] *Department of Physics, The University of Tokyo, Tokyo, Japan*

[2] *Physics & Informatics Laboratories, NTT Research, Inc., Sunnyvale, CA, USA*

[3] *Center for Brain Science, Harvard University, Cambridge, USA*

[4] *EECS Department, University of Michigan, Ann Arbor, USA*

[5] *Institute for Physics of Intelligence, The University of Tokyo, 7-3-1 Hongo, Bunkyo-ku, Tokyo*

*1* [6] *RIKEN Center for Emergent Matter Science (CEMS), Wako, Saitama, Japan*

**Editors:** Sophia Sanborn, Christian Shewmake, Simone Azeglio, Arianna Di Bernardo, Nina Miolane

## Abstract

We derive an analytically tractable theory of SSL landscape and show that it accurately captures an array of collapse phenomena and identifies their causes.

Self-supervised learning (SSL) methods have achieved remarkable results in learning good representations without labeled data. SSL loss functions are designed to promote representational similarity between pairs of related samples while using explicit penalties or asymmetric dynamics to ensure that the distance between unrelated samples remains large. In practice, however, SSL training often experiences the failure mode of *dimensional collapse* (Jing et al., 2021; Tian et al., 2021; Pokle et al., 2022), where the learned representation spans a low dimensional subspace of the overall available space. In the extreme case, this failure mode instantiates as a *complete collapse*, where the learned representation becomes zero-rank, and no informative features can be extracted. Additional related works are discussed in Appendix A. In this work, we analytically solve the effective landscapes of linear models trained on several popular losses used in self-supervised learning. Combining theory and empirical results, a key insight we offer is: *collapses of representations are strongly dependent on the stability of the last layer at the origin and happens when a broken symmetry is restored.*

## 1. A Landscape Theory of Self-Supervised-Learning

Let $\{\hat{x}_i\}_i^N$ be a dataset with $N$ data points. For two data points $\hat{x}$ and $\hat{chi}$, we augment it with an i.i.d. noise $\epsilon$ such that $x := \hat{x} + \epsilon$ and $\chi := \hat{\chi} + \epsilon'$ are independently sampled data with independent augmentations. To be concrete, we start with considering the standard contrastive loss, InfoNCE (Oord et al., 2018):

$$L = \mathbb{E}_\epsilon \left[ -\sum_{i=1}^N \log \frac{\exp(-|f(x_i) - f(x_i')|^2/2)}{\sum_{j \neq i} \exp(-|f(x_i) - f(\chi_j)|^2/2) + \exp(-|f(x_i) - f(x_i')|^2/2)} \right], \qquad (1)$$

where $f(x) \in \mathbb{R}^{d_1}$ is the model output; all $x$, $x'$ and $\chi$ are augmented data points for some independent additive noise $\epsilon$ such that $\mathbb{E}_\epsilon[x] = \hat{x} = \mathbb{E}_\epsilon[x'] \neq \mathbb{E}_\epsilon[\chi] = \hat{\chi}$. We decompose the model output into a general function $\phi(x) \in \mathbb{R}^{d_0}$ and the last-layer weight matrix $W \in \mathbb{R}^{d_1 \times d_0}$: $f(x) = W\phi(x)$. The covariance of $\phi(\hat{x})$ is $A_0 := \mathbb{E}_{\hat{x}}[\phi(\hat{x})\phi(\hat{x})^T]$, and the covariance of the data-augmented penultimate layer representation is $\Sigma := \mathbb{E}_x[\phi(x)\phi(x)^T]$. The effect of data augmentation on the learned representation is captured through a symmetric matrix $C := \Sigma - A_0$. For a general $\phi$, the eigenvalues of $C$ can be either positive or negative. When $\phi$ is the identity mapping, $A_0$ becomes the empirical data covariance, $C$ becomes PSD and is the covariance of the noise $\epsilon$, and $\Sigma$ is the covariance of the augmented data. In some sense, this loss function captures the essence of SSL: the numerator encourages the representation $f(x)$ to be closer to the representation of similar data, and the denominator encourages a separation between dissimilar data.

For a fixed set of noises, we can write the InfoNCE in a cleaner form:

$$L_\epsilon = \mathbb{E}_{\hat{x}}\left\{\frac{1}{2}|f(x) - f(x')|^2 + \log \mathbb{E}_{\hat{\chi}}\left[\exp\left(-\frac{1}{2}|f(x) - f(\chi)|^2\right)\right]\right\}, \tag{2}$$

where we used $\mathbb{E}_{\hat{x}}$ to denote an averaging over the training set. In this notation, we have $\mathbb{E}_\epsilon \mathbb{E}_{\hat{x}}[x] = \mathbb{E}_x[x]$ and $\mathbb{E}_\epsilon[L_\epsilon] = L$. For a quantitative understanding, we mainly focus on the case when $\phi$ is the identity function. We discuss the general nonlinear case in Section 1.3. The proofs are presented in Appendix E.

## 1.1. Landscape of a Linear Model

**NT-xent**. As in Tian (2022), we note InfoNCE can be generalized as follows:

$$L = \mathbb{E}_\epsilon\left[-\sum_{i=1}^{N} \log \frac{\exp(-|f(x_i) - f(x_i')|^2/2)}{\sum_{\chi \neq x} \exp(-|f(x_i) - f(\chi_j)|^2/2) + \alpha \exp(-|f(x_i) - f(x_i')|^2/2)}\right]. \tag{3}$$

Different from InfoNCE, one term in the denominator is weighted by a factor $\alpha \geq 0$. Two interesting limits are $\alpha = 1$, where we recover the InfoNCE loss, and $\alpha = 0$, where we obtain the popular NT-xent loss used in SimCLR (Chen et al., 2020). For general $\alpha$, we refer to this loss as the *weighted InfoNCE*. For a perceptron, the leading terms of the loss function is $L = \frac{1-\alpha}{N}\mathrm{Tr}[WCW^T] - \mathrm{Tr}[WA_0W^T] + \frac{1}{8}\mathrm{Var}[|W(x - \chi)|^2]$.

In fact, for the losses functions we consider, the leading order terms of the loss function all take the following rather universal form, for some symmetric matrix $B$,

$$L = -\mathrm{Tr}[WBW^T] + \frac{1}{8}\mathrm{Var}[|W(x - \chi)|^2]. \tag{4}$$

**Landscape Analysis**. When training ends, one expects the model to locate at (at least close to) a stationary point of the loss. It is thus important to identify all the stationary points of this loss function.

**Theorem 1** *Let $d^* := \min(d_0, d_1)$. Let the data and noise be Gaussian. All stationary points $W$ of Eq. (4) satisfy $W^TW = \frac{1}{2}\Sigma^{-1/2}UM\Lambda U^T\Sigma^{-1/2}$, where $U\Lambda U^T$ is the eigenvalue decomposition of $\Sigma^{-1/2}B\Sigma^{-1/2}$, and $M$ is an arbitrary (masking) diagonal matrix containing only zero or one such that (1) $M_{ii} = 0$ if $\Lambda_{ii} < 0$ and (2) contain at most $d^*$ nonzero terms.*

*Additionally, if $C$ and $A_0$ commute, all stationary points satisfy $W^TW = \frac{1}{2}\Sigma^{-1}B_M\Sigma^{-1}$, where $B_M$ denotes the matrix obtained by masking the eigenvalues of $B$ with $M$.*

## Extended Abstract Track

This stationary-point condition implies the direct cause of the dimensional collapse. Namely, dimensional collapse happens when the eigenvalues of the matrix $B$ become negative. The eigenvalues of $B$, in turn, depend on the competition between data augmentation and the data feature. Comparing the commuting case with the noncommuting case, we see that the main difference is that when $CA_0 \neq A_0C$, the augmentation can also change the orientation of the learned representation; otherwise, augmentation only affects the eigenvalues. To focus on the most important terms, we now assume that the augmentation is well-aligned with the features such that the augmentation covariance commute with the data covariance. From now on, we assume $CA_0 = A_0C$.

For the case of weighted InfoNCE, we have that $B = A_0 - \frac{1-\alpha}{N}C$. Let $a_i$ denote the $i$-th eigenvalue of the $A$ and $c_i$ that of $C$ viewed in a predetermined order; then, the $i$th subspace collapses when $\frac{1-\alpha}{N}c_i \geq a_i$, namely, when the variation introduced by the noise dominates that of the original data. Importantly, this collapse is a property shared by *all* stationary points of the landscape, and one cannot hope to fix the problem by, say, biasing the gradient descent towards a certain type of local minima. When weight decay is used, the condition for collapse becomes $\frac{1-\alpha}{N}c_i + \gamma \geq a_i$. It becomes easier to cause a collapse when weight decay is used.

Because the stationary points contain collapsed solutions where the eigenvalues of $W^TW$ are zero, one is naturally interested in how likely it is to converge to these solutions. The following proposition implies that the loss landscape of contrastive SSL (with a linear model) is rather benign because all local minima must achieve a maximum possible rank.

**Proposition 2** ($W^TW$ achieves maximum possible rank) *Let $m$ denote the number of positive eigenvalues $B$. Then,* $\mathrm{rank}(W^TW) = \min(m, d^*)$ *for any local minimum.*

### 1.2. Landscape with Normalization

It is common in practice to normalize the learned representation such that $\|f(x)\|^2 = c$. When the normalization is applied, only the direction of the learned representation matters. While this is a simple trick in practice, its implication on the landscape is poorly understood. In this section, we extend our theory to analyze the effect of normalization.

We model the effect of normalization as a regularization term: $R := (\mathbb{E}_x\|f(x)\|^2 - c)^2$:

$$L = Eq.~(4) + \kappa R. \tag{5}$$

This regularization term achieves two things: (1) $\|f(x)\|^2 = c$ is a minimizer of the loss function; (2) the regularization is invariant to a rotation of the representation. This loss function can also be seen as a mathematical model of the VICReg loss (Bardes et al., 2021), where $R$ effectively models the variance regularization term of VICReg loss and $\kappa$ is its strength. This modeling is necessary because the variance term of the original VICReg is not differentiable and thus cannot be expanded. The proposed term $R$ captures the essence of the variance term because it also encourages the representation to have a constant variance. Our theory also explains why the VICReg is observed to experience collapses when $\kappa$ is not large enough. As $\kappa$ tends to infinity, this constraint will become perfectly satisfied. We thus take the infinite $\kappa$ limit to study the effect of normalization.

The following proposition gives a condition that all stationary points of Eq. (5) satisfy.

**Proposition 3** *Let $\rho(W) := \mathrm{Tr}[W\Sigma W^T]$, $B' := B + 2\kappa(c-\rho)\Sigma$, and let $\Lambda_i$ be the eigenvalues of $B'$. Then, every stationary point of Eq. (5) satisfy $W^TW = \frac{1}{2}\Sigma^{-1}B'_M\Sigma^{-1}$, where $M$ is an*

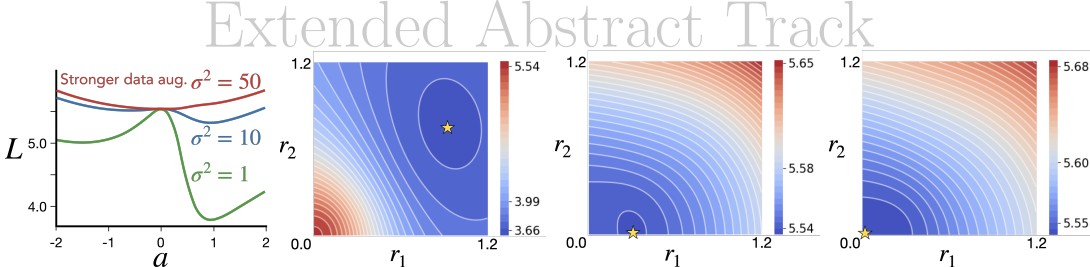

Figure 1: Landscape of Resnet18 on CIFAR10 with SimCLR. (a) Training objective $L$ as a function of a rescaling of the last layer $W \to aW$. The origin becomes a local minimum as the data augmentation $\sigma^2$ gets stronger. (b-d) $L$ as a function of a $2d$ rescaling of the last layer where the data augmentation strength is (b) small (no collapse), (c) intermediate (dimensional collapse), and (d) strong (complete collapse). Use of data augmentation changes the stability of the origin, a qualitative change that leads to different types of collapses.

*arbitrary diagonal mask of the eigenvalues of $B'$ containing only zero or one such that (1) $M_{ii} = 0$ if $\Lambda_i < 0$ and (2) contain at most $d^*$ nonzero terms.*

Compared with the unnormalized case, the term $2\kappa(1 - \rho)\Sigma_M$ emerges due to normalization. The effect of normalization is as expected: it shrinks the norm of the model if $\rho > 1$, and it expands the model if $\rho < 1$, and it does not have any effect if we have already achieved $\rho = 1$. Interestingly, this rescaling effect is anisotropic and stronger along the directions of larger eigenvalues of the covariance of the augmented data $\Sigma$. Section D.3 directly finds the solution of $\rho$. For a finite $\kappa$, these results suggest that collapses can still happen. For VICReg, $B = -A_0$, and the complete collapse can happen when $\kappa \ll \|A_0\|/c\|\Sigma\|$ – this explains the experimental observation of collapses for small values of $\kappa$ in (Bardes et al., 2021). Lastly, to understand normalization, we are interested in the case of $\kappa \to \infty$. We discuss this case in detail in Appendix D.1. Our result can also be applied to analyze spectral contrastive loss (SCL) (HaoChen et al., 2021) and Barlow Twins, which are discussed in Appendix D.4

### 1.3. Relevance to Nonlinear Models

An important question is how much the analysis connects to deep nonlinear models. In fact, the loss landscape we have studied is close to the most general landscape one can have. Let $L(f(x))$ be a general SSL loss function for data point $x$. The quality of the learned representation should be independent of the population-level orientation of the representation. Therefore, the loss function should be rotationally invariant: for any rotation matrix $R$, $L(x) = L(Rf(x))$. This invariance implies that the loss expands as $L(f(x)) = af(x)^T f(x) + b[f(x)^T f(x)]^2 + O(f(x)^6)$. Note that all the odd-order terms of $f(x)$ vanish due to the rotational symmetry. Substituting $f(x) = W\phi(x)$ in the loss function, we obtain the general form of landscape that $W$ obeys:

$$L(W, \phi) = \text{Tr}[W^T W A(\phi)] + \sum_{ijklmn} W_{im} W_{jm} W_{kn} W_{ln} Z_{ijki}(\phi), \tag{6}$$

where $A$ and $Z$ are dependent on $\phi$. All the examples we have studied take this form. For $W$, its collapse depends on the stability of the matrix $A$. Thus the study of the stability of the matrix $A$ is crucial for our understanding. To illustrate, we train a Resnet18 on CIFAR10 with the SimCLR loss with normalization and with weight decay strength $10^{-3}$ until convergence to obtain the converged weights $W^*$. We inject independent Gaussian noises with variance $\sigma^2$ as data augmentation. The representation has a dimension 128. We

# Extended Abstract Track

rescale the weight matrix of the last layer $W_{\text{last}}^*$ by a factor of $a$ and compute the loss as a function of $a$. See Figure 1-a. We then partition the singular values of $W_{\text{last}}^*$ into the larger and smaller half. We rescale the larger half by a factor $r_1$ and the smaller half by $r_2$. We plot the loss as a $2d$ function of $(r_1, r_2)$ in Figure 1. See Appendix B for more experiments that validate our theory on both linear and nonlinear models.

Those familiar with statistical physics should note that the proposed theory mimics the Landau theory of second-order phase transitions. When treating the loss function as the free energy, the square root of the eigenvalues $\sqrt{\lambda}$ of $W^T W$ are the order parameters of the system, and the phase transitions happen when $\lambda$ turns from 0 to positive. These transitions (collapses) happen because of *symmetry breaking*: the loss function (2) is symmetric in the sign of $W$. Yet, for any nontrivial learning, $W$ must be nonzero. A symmetry breaking of the sign of $W$ needs to happen for learning. This phase transition phenomenon with the 0.5 scaling is also in line with the neural collapse phenomenon in supervised learning (Ziyin and Ueda, 2022).

## 2. Conclusion

In this work, we approached the problem of collapses in SSL from a landscape and symmetry breaking perspective. We analytically solved an effective landscape that can be extended to understand the effect of normalization. Our result suggests that dimensional collapse can be well understood in the minimal setting and is something neutral to learning on its own. We showed that when task-irrelevant dimensions are targeted, dimensional collapse can result in dramatically improved performance, whereas an uninformative noise will (without good luck) leads to collapses in the dimensions that are relevant to the task. It is thus important for practitioners to devise targeted data augmentation mechanisms that incorporate the correct domain knowledge. The proposed theory can serve as a theoretical foundation and baseline of any advanced theory of collapses because a correct theory should agree with our results when restricting to the case of a linear model. We advocated the thesis that the local geometry of the loss landscape around the origin is an essential component for understanding collapses, and this should invite more future work to understand the landscape around the origin.

The limitation of our work is clear; our result only identifies the causes of the collapse that can be directly attributed to the low-rank structure of the local minima of the landscape. One possible alternative cause of the collapse is dynamics. For example, having a large learning rate and small batch can sometimes cause a convergence towards the saddle points in the landscape (Ziyin et al., 2021), which, as we have shown, are the collapsed solutions. Investigating the role of dynamics in the collapse is thus a crucial future problem.

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

# Extended Abstract Track

Jordan T Ash, Surbhi Goel, Akshay Krishnamurthy, and Dipendra Misra. Investigating the role of negatives in contrastive representation learning. *Int. Conf. on Artificial Intelligence and Statistics*, 2021.

Adrien Bardes, Jean Ponce, and Yann LeCun. Vicreg: Variance-invariance-covariance regularization for self-supervised learning. *arXiv preprint arXiv:2105.04906*, 2021.

Ting Chen, Simon Kornblith, Mohammad Norouzi, and Geoffrey Hinton. A simple framework for contrastive learning of visual representations. In *International conference on machine learning*, pages 1597–1607. PMLR, 2020.

Xinlei Chen and Kaiming He. Exploring Simple Siamese Representation Learning. In *Proc. Int. Conf. on Computer Vision and Pattern Recognition (CVPR)*, 2021.

Romain Cosentino, Anirvan Sengupta, Salman Avestimehr, Mahdi Soltanolkotabi, Antonio Ortega, Ted Willke, and Mariano Tepper. Toward a geometrical understanding of self-supervised contrastive learning. *arXiv preprint arXiv:2205.06926*, 2022.

Aleksandr Ermolov, Aliaksandr Siarohin, Enver Sangineto, and Nicu Sebe. Whitening for self-supervised representation learning. In *International Conference on Machine Learning*, pages 3015–3024. PMLR, 2021.

Jean-Bastien Grill, Florian Strub, Florent Altché, Corentin Tallec, Pierre Richemond, Elena Buchatskaya, Carl Doersch, Bernardo Avila Pires, Zhaohan Guo, Mohammad Gheshlaghi Azar, Bilal Piot, koray kavukcuoglu, Remi Munos, and Michal Valko. Bootstrap your own latent: A new approach to self-supervised Learning. In *Proc. Adv. on Neural Information Processing Systems (NeurIPS)*, 2020.

Jeff Z HaoChen, Colin Wei, Adrien Gaidon, and Tengyu Ma. Provable guarantees for self-supervised deep learning with spectral contrastive loss. *Advances in Neural Information Processing Systems*, 34:5000–5011, 2021.

Tianyu Hua, Wenxiao Wang, Zihui Xue, Sucheng Ren, Yue Wang, and Hang Zhao. On feature decorrelation in self-supervised learning. In *Proceedings of the IEEE/CVF International Conference on Computer Vision*, pages 9598–9608, 2021.

Li Jing, Pascal Vincent, Yann LeCun, and Yuandong Tian. Understanding dimensional collapse in contrastive self-supervised learning. *arXiv preprint arXiv:2110.09348*, 2021.

Kenji Kawaguchi. Deep learning without poor local minima. *Advances in Neural Information Processing Systems*, 29:586–594, 2016.

Julius Kugelgen, Yash Sharma, Luigi Gresle, Wieland Brendel, Bernhard Scholkopf, Michel Besserve, and Francesco Locatello. Self-Supervised Learning with Data Augmentations Provably Isolates Content from Style. *arXiv*, abs/2106.04619, 2021.

Hong Liu, Jeff Z HaoChen, Adrien Gaidon, and Tengyu Ma. Self-supervised learning is more robust to dataset imbalance. *International Conference on Learning Representations*, 2021.

James Lucas, George Tucker, Roger Grosse, and Mohammad Norouzi. Don't blame the elbo! a linear vae perspective on posterior collapse, 2019.

Jovana Mitrovic, Brian McWilliams, Jacob Walker, Lars Buesing, and Charles Blundell. Representation learning via invariant causal mechanisms. *arXiv preprint arXiv:2010.07922*, 2020.

Kento Nozawa and Issei Sato. Understanding negative samples in instance discriminative self-supervised representation learning. *Advances in Neural Information Processing Systems*, 34:5784–5797, 2021.

Aaron van den Oord, Yazhe Li, and Oriol Vinyals. Representation learning with contrastive predictive coding. *arXiv preprint arXiv:1807.03748*, 2018.

Ashwini Pokle, Jinjin Tian, Yuchen Li, and Andrej Risteski. Contrasting the landscape of contrastive and non-contrastive learning. *arXiv preprint arXiv:2203.15702*, 2022.

Joshua Robinson, Li Sun, Ke Yu, Kayhan Batmanghelich, Stefanie Jegelka, and Suvrit Sra. Can contrastive learning avoid shortcut solutions? *Advances in neural information processing systems*, 34:4974–4986, 2021.

Nikunj Saunshi, Jordan Ash, Surbhi Goel, Dipendra Misra, Cyril Zhang, Sanjeev Arora, Sham Kakade, and Akshay Krishnamurthy. Understanding Contrastive Learning Requires Incorporating Inductive Biases. In *Proc. Int. Conf. on Machine Learning (ICML)*, 2022.

Andrew M Saxe, James L McClelland, and Surya Ganguli. Exact solutions to the nonlinear dynamics of learning in deep linear neural networks. *arXiv preprint arXiv:1312.6120*, 2013.

Yonglong Tian, Chen Sun, Ben Poole, Dilip Krishnan, Cordelia Schmid, and Phillip Isola. What makes for good views for contrastive learning? *Advances in Neural Information Processing Systems*, 33:6827–6839, 2020.

Yuandong Tian. Deep contrastive learning is provably (almost) principal component analysis. *arXiv preprint arXiv:2201.12680*, 2022.

Yuandong Tian, Xinlei Chen, and Surya Ganguli. Understanding self-supervised Learning Dynamics without Contrastive Pairs. In *Proc. Int. Conf. on Machine Learning (ICML)*, 2021.

Christopher Tosh, Akshay Krishnamurthy, and Daniel Hsu. Contrastive estimation reveals topic posterior information to linear models. *J. Mach. Learn. Res.*, 22:281–1, 2021a.

Christopher Tosh, Akshay Krishnamurthy, and Daniel Hsu. Contrastive learning, multi-view redundancy, and linear models. In *Algorithmic Learning Theory*, pages 1179–1206. PMLR, 2021b.

Puja Trivedi, Ekdeep Singh Lubana, Mark Heimann, Danai Koutra, and Jayaraman J Thiagarajan. Analyzing data-centric properties for contrastive learning on graphs. *arXiv preprint arXiv:2208.02810*, 2022.

# Extended Abstract Track

Yao-Hung Tsai, Yue Wu, Ruslan Salakhutdinov, and Louis-Philippe Morency. Self-supervised Learning from a Multi-view Perspective. In *Proc. Int. Conf. on Learning Representations (ICLR)*, 2021a.

Yao-Hung Hubert Tsai, Martin Q Ma, Muqiao Yang, Han Zhao, Louis-Philippe Morency, and Ruslan Salakhutdinov. Self-supervised representation learning with relative predictive coding. *International Conference on Learning Representations*, 2021b.

Feng Wang and Huaping Liu. Understanding the behaviour of contrastive loss. In *Proceedings of the IEEE/CVF conference on computer vision and pattern recognition*, pages 2495–2504, 2021.

Tongzhou Wang and Philip Isola. Understanding Contrastive Representation Learning through Alignment and Uniformity on the Hypersphere. In *Proc. Int. Conf. on Machine Learning (ICML)*, 2020.

Yifei Wang, Qi Zhang, Yisen Wang, Jiansheng Yang, and Zhouchen Lin. Chaos is a ladder: A new theoretical understanding of contrastive learning via augmentation overlap. *International Conference on Learning Representations*, 2022.

Zihao Wang and Liu Ziyin. Posterior collapse of a linear latent variable model. *arXiv preprint arXiv:2205.04009*, 2022.

Colin Wei, Kendrick Shen, Yining Chen, and Tengyu Ma. Theoretical Analysis of Self-Training with Deep Networks on Unlabeled Data. In *Proc. Int. Conf. on Learning Representations (ICLR)*, 2021.

Jure Zbontar, Li Jing, Ishan Misra, Yann LeCun, and Stéphane Deny. Barlow twins: Self-supervised learning via redundancy reduction. In *International Conference on Machine Learning*, pages 12310–12320. PMLR, 2021.

Roland Zimmerman, Yash Sharma, Steffen Schneider, Matthias Bethge, and Wieland Brendel. Contrastive Learning Inverts the Data Generating Process. In *Proc. Int. Conf. on Machine Learning (ICML)*, 2021.

Liu Ziyin and Masahito Ueda. Exact phase transitions in deep learning. *arXiv preprint arXiv:2205.12510*, 2022.

Liu Ziyin, Botao Li, James B Simon, and Masahito Ueda. Sgd with a constant large learning rate can converge to local maxima. *arXiv preprint arXiv:2107.11774*, 2021.

Extended Abstract Track

## Appendix A. Related Works

**SSL and Collapses.** Prior literature has often argued collapse as a harmful phenomenon that can deteriorate downstream task performance (Jing et al., 2021; Zbontar et al., 2021). Preventing such collapsed representations is a frequently discussed topic in literature (Hua et al., 2021; Jing et al., 2021; Pokle et al., 2022; Tian et al., 2021) and has motivated the design of several SSL techniques (Zbontar et al., 2021; Bardes et al., 2021; Ermolov et al., 2021). However, in direct contrast, Cosentino et al. (2022) empirically showed that dimensional collapses under strong augmentations could dramatically improve generalization performance. Our work demystifies these conflicting results by finding analytic solutions to loss landscapes of several standard SSL techniques.

**Theoretical Advances in SSL.** Recently, several advances have been made towards understanding the success of SSL techniques from different perspectives: e.g., learning theory (Arora et al., 2019; HaoChen et al., 2021; Saunshi et al., 2022; Nozawa and Sato, 2021; Ash et al., 2021; Wei et al., 2021), information theory (Tsai et al., 2021a,b; Tosh et al., 2021a,b), causality and data-generating processes (Zimmerman et al., 2021; Kugelgen et al., 2021; Trivedi et al., 2022; Tian et al., 2020; Mitrovic et al., 2020; Wang et al., 2022), dynamics (Wang and Isola, 2020; Tian et al., 2021; Tian, 2022; Wang and Liu, 2021), and loss landscapes (Jing et al., 2021; Pokle et al., 2022). These advances have unveiled practically useful properties of SSL, such as robustness to dataset imbalance (Liu et al., 2021) and principled solutions to avoid spurious correlations (Robinson et al., 2021).

The work by Jing et al. (2021) is the closest to ours in problem setting. In that paper, the authors focused on studying the linearized learning dynamics and suggested that a competition between the feature signal strength and augmentation strength can lead to dimensional collapse. In contrast, our focus is on the landscape and our result implies that this feature-augmentation competition on its own is insufficient to cause a dimensional collapse. In fact, we show that there will be no collapse in the setting studied by Jing et al. (2021).

**Interpretability of linear models.** Previous works have demonstrated how linear models are often sufficient to reproduce phenomena observed in non-linear deep networks (Saxe et al., 2013; Kawaguchi, 2016). If an observed phenomenon does not occur for linear models, one can conclude that the use of nonlinearity is a necessary condition for this phenomenon. If a phenomenon happens for both models, one naturally concludes that non-linearity is not necessary to cause the phenomena. For example, posterior collapses in Bayesian deep learning were first thought to be caused by expressivity due to non-linearity (Alemi et al., 2018); however, recent works (Lucas et al., 2019; Wang and Ziyin, 2022) found that models without non-linearity can still induce the collapse.

## Appendix B. Additional Numerical Results

In this section, we validate our theory with numerical results. Unless specified otherwise, the dimension of the learned representation is set to be equal to the input dimension: $d_0 = d_1$.

**No Collapse for InfoNCE**. We showed that there is no collapse at all for the vanilla InfoNCE, no matter how strong the augmentation is. Our result implies that the smallest singular of the model $W$ scales as $\sigma^4$ where $\sigma^2$ is the strength (namely, the variance) of the augmentation. See the left panel of Fig. 2. We use the vanilla InfoNCE loss defined in (1)

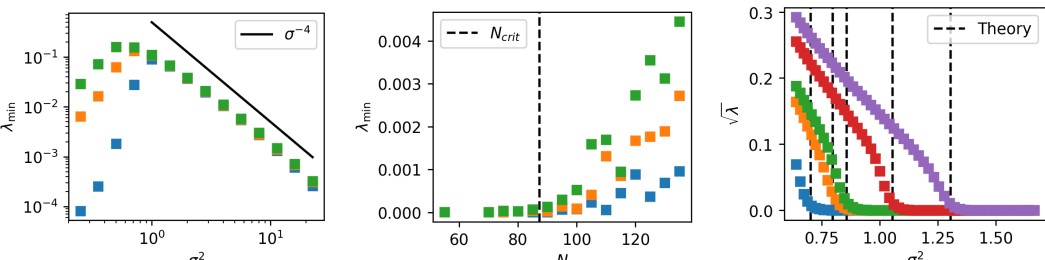

Figure 2: The three smallest singular values of $W^T W$ as a function of the augmentation strength. We see that our effective landscape theory around the origin accurately captures collapses in learning. **Left**: Vanilla InfoNCE . As the theory suggests, the singular values scale as $\sigma^4$ and do not vanish for any finite value of $\sigma$. **Mid**: Weight InfoNCE. $\alpha = 0.1$, $\sigma = 5$. Collapse happens at the critical dataset size predicted by the theory. **Right**: (Sqrt) Eigenvalues of $WW^T$ in $\beta$-InfoNCE. The collapses can be well controlled.

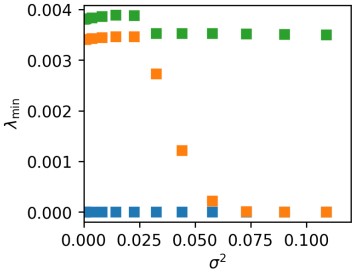

Figure 3: A collapse happens easily when the learned representation is normalized. The smallest eigenvalues of $A_0$ are roughly 0.2, and the collapse happens much before the noise reaches this strength.

with a linear model. The training set is sampled from $\mathcal{N}(0, I_{32})$. The training proceeds with Adam with a learning rate of $6e - 4$ with full batch training for 5000 iterations. We use a simple diagonal Gaussian noise with variance $\sigma^2$ for data augmentation. We see that the singular values scale as $\sigma^4$ and never vanishes, as the theory predicts.

**Nonrobust Collapses of Weighted InfoNCE**. We now demonstrate that, as the theory predicts, collapses of weighted InfoNCE depend strongly on the dataset size. We use the same dataset and training procedure as the previous experiment. We set $\alpha = 0.1$ and change the size of the training set. Theory suggests that for a collapse in the $i-$th subspace to happen, the size of the dataset needs to obey

$$N > \frac{a_i}{c_i(1 - \alpha)} := N_{crit}. \tag{7}$$

See the middle panel of Figure 2. We show the smallest three eigenvalues of $W^T W$ (roughly having similar magnitudes), and the critical dataset size for the smallest eigenvalue. We see that the theoretical threshold of collapse agrees well with where the collapse actually happens.

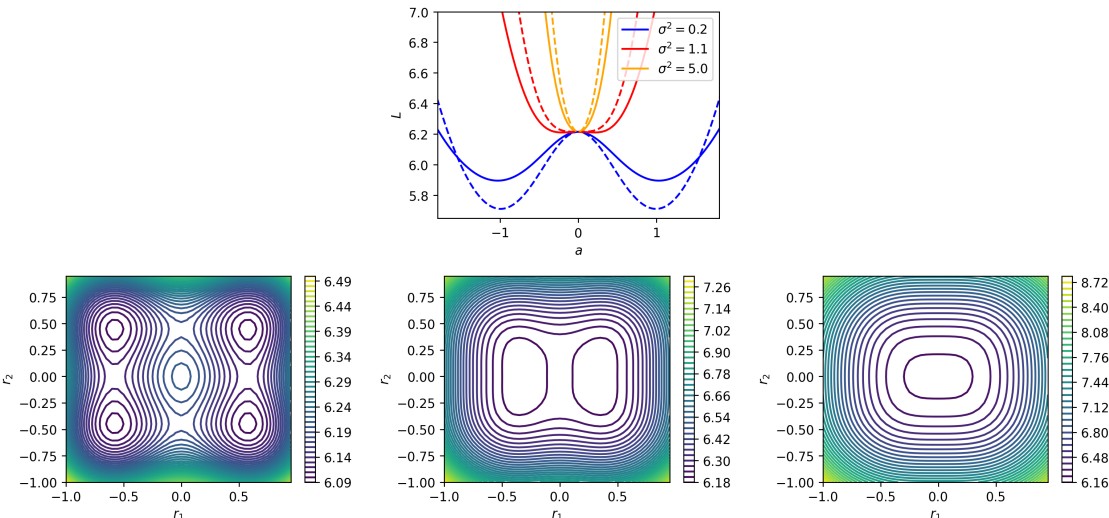

Figure 4: The Landscape of nonlinear models is very similar to the landscape of linear models. **Top**: $1d$ projection of the landscape of a two-layer tanh and ReLU network. **Bottom Left**: the landscape of a 2D projection of the last layer of a nonlinear model with a weak augmentation. **Middle**: with intermediate augmentation. **Right**: with strong augmentation.

**Collapses in $\beta$-InfoNCE**. With $\beta < 1$, one can cause collapses in a predictable and controllable way. In this experiment, we let $d_0 = 5$ and we plot all five eigenvalues of $W^T W$ as we increase the strength of an isotropic augmentation. As the numerical results show, collapses happen at the points predicted by the theory.

**Normalization Causes Dimensional Collapse**. We also plot the three smallest eigenvalues of $W^T W$ when we apply the standard representation normalization in practice: $f(x) \rightarrow f(x)/\|f(x)\|$. To facilitate comparison, we also use the same dataset and training procedure as before. See Figure 3. We see that normalization does cause a collapse in the smallest eigenvalues at an augmentation strength much smaller than the feature variation.

## Appendix C. Landscape of a Nonlinear Model

In this section, we plot the landscape of the layer of nonlinear models on the same synthetic dataset we outlined in the previous section. We train a three-layer nonlinear network with output dimension 2 with SGD until convergence. We then rescale the optimized weight of the last by a factor $a$: $W_{last} \rightarrow a W_{last}$ and plot the loss function along this direction. See the top panel of Figure 4 for both the tanh and the ReLU nonlinearity. We then rescale the two rows of the weight matrix of the model by $r_1$ and $r_2$ respectively: $W = (w_1, W_2)^T \rightarrow (r_1 w_1, r_2 w_2)$.

# Extended Abstract Track

## Appendix D. Additional Theoretical Concerns

### D.1. Collapse condition for normalization

Combining Proposition 3 and 5, one obtains $\lim_{\kappa \to \infty} W_\kappa^T W_\kappa = \frac{1}{2}\Sigma^{-1}\left[B_M + \frac{2c - \text{Tr}[\Sigma^{-1}B_M]}{d_M}\Sigma_M\right]\Sigma^{-1}$. Because the eigenvalues of $WW^T$ must be positive, the following condition holds for all solutions:

$$\lambda_i + 2c/d_M > \bar{\lambda}. \tag{8}$$

where $\lambda_i$ are the eigenvalues of $\Sigma^{-1}B_M$ and $\bar{\lambda}$ is its average. Namely, for the $i$-th dimension not to collapse, it must be smaller than the average eigenvalues by at most $2c/d_M$. Any smaller eigenvalues must collapse. Compared to the case without normalization, normalization makes collapses dependent on the *relative* strength of each feature and augmentation. One finds that the condition for collapse becomes heavily dependent on the data structure, and there are cases where collapses become harder, and there are cases where collapses become much easier. Importantly, it also becomes the case that a sufficiently strong augmentation can always cause a collapse in the corresponding subspace.

The important condition for collapse in Eq. (8) can be better understood by considering the extreme cases. First of all, note that the eigenvalues of $\Sigma B_M$ are bounded between $-1$ and $1$

$$-1 \le \frac{a_i - c_i}{a_i + c_i} \le 1, \tag{9}$$

and $-1$ is achieved when $c_i \gg a_i$, and $1$ is achieved when $a_i \gg c_i$.

When the augmentation is negligibly small, $\Sigma^{-1}B_M \approx M$, and $\lambda_i \approx \bar{\lambda} = 1$, the condition thus becomes

$$\frac{2}{d_M} > 0, \tag{10}$$

which always holds. Thus, a sufficiently small augmentation will never cause collapse. Next, when we apply very strong augmentation to the $j$-th subspace and zero augmentation to the others, the condition for the non-augmented spaces becomes

$$1 + \frac{2}{d_M} > \frac{d_M - 2}{d_M}, \tag{11}$$

meaning that the collapse will not happen. For the $j$-th space, the condition is

$$-1 + \frac{2}{d_M} > \frac{d_M - 2}{d_M} (\Longleftrightarrow) \frac{4}{d_M} > 2, \tag{12}$$

which is only possible when $d_M = 1$, namely, the strongly augmented space is the only space that does not collapse. This is reasonable when the original data is rank-1 because the normalization will ensure that this space does not collapse, but when the original data is not rank-1, this stationary point will be a saddle and will not be preferred by gradient descent. In different word, a strong enough augmentation will cause a collapse in the corresponding subspace, as is the case without normalization.

It is also interesting to note that having $c_i \ge a_i$ is no longer sufficient to cause a collapse. For example, let $c_1 = 0$ and $c_j = a_j$ for $j \ne 1$. The condition for $j \ne 1$ becomes

$$\frac{2}{d_M} > \frac{1}{d_M}, \tag{13}$$

# Extended Abstract Track

which always holds. At the same time, it does not mean that collapsing has become harder in general. For example, it is also possible for $c_i < a_i$ to cause a collapse. Suppose we add a weak augmentation only to the first subspace such that $a_i - c_i = \epsilon > 0$, the condition for this dimension to not to collapse is

$$\frac{\epsilon}{a_i + c_i} + \frac{2}{d_M} > \frac{d_M - 1 + \epsilon}{d_M}, \tag{14}$$

which can be violated whenever $\epsilon < \frac{(a_i + c_i)(d_M - 3)}{a_i + c_i + d_m}$. Namely, in some cases, normalization can in fact facilitate collapse.

## D.2. Effect of Bias

**Effect of Bias**. Lastly, we study the effect of explicitly having a bias term: $Wx \to Wx + b$. First of all, when there is no normalization, the bias term does not affect the solution because the loss landscape is invariant to a translation in the learned representation. However, this effect dramatically changes if we apply normalization at the same time. This is because normalization removes the translation symmetry of the effective loss, and the trivial solution $W = 0$, $b = 1$ becomes the simplest way to achieve the norm−1 constraint. Our result shows that the addition of bias dramatically affects the stationary points.

**Theorem 4** *Let $f(x) = Wx + b$ and $\mathbb{E}[x] = 0$. Then, all stationary points satisfy $W^T W = \frac{1}{2}\Sigma^{-1} B_M \Sigma^{-1}$, subject to the constraint that $\rho(W) = \text{Tr}[W^T \Sigma W] \leq c$.*

Namely, the solution reverts to the case where there is no normalization at all, except that the norm of the solution can no longer be larger than $c$. This upper bound can make collapses much easier to happen. For example, if $c < (a_i - c_i)/(a_i + c_i)$ for all $i$, a complete collapse can happen despite normalization. When $c = 1$ and $c_i \ll a_i$, $\rho \approx d_M/2$ and the constraint indicates that $d_M \leq 2$: when the augmentation is very weak, there are at most 2 nontrivial subspaces. This is too restrictive for learning a meaningful representation, which helps us understand why dimensional collapse can harm learning in practice. The fact that simple normalization cannot prevent collapse has been noticed for a while for the simplest case of a cosine-similarity loss, and our result explains why previous works have tried to introduce asymmetry to cosine similarity to avoid collapses (Grill et al., 2020; Chen and He, 2021).

## D.3. Solution of $\rho$

The next theorem gives the explicit form of $\rho$ at the stationary points.

**Proposition 5** *For any stationary point $W^*$,*

$$c - \rho(W^*) = \frac{c - \frac{1}{2}\text{Tr}[\Sigma^{-1} B_M]}{1 + \kappa d_M}, \tag{15}$$

*where $d_M$ is the number of non-zero eigenvalues of $B_M'$.*

### D.4. Relevant Loss Functions

Having developed a framework for understanding normalization, we show that other common loss functions in SSL can also be written in the form given in Eq. (4). The spectral contrastive loss (SCL) (HaoChen et al., 2021) reads

$$L_{SCL} = -2\mathbb{E}[f(x)^T f(x')]) + \mathbb{E}[(f(x)^T f(\chi))^2] + const. \qquad \text{s.t. } \|f(x)\|^2 = 1. \qquad (16)$$

Let $f(x) = Wx$ be linear, the distributions are zero-mean Gaussian, and ignore the normalization. This loss function becomes

$$L_{SCL} = -2\mathrm{Tr}[WCW^T] + \mathrm{Tr}[W\Sigma W^T W\Sigma W^T]. \qquad (17)$$

When normalization exists, we can apply the result in Section 1.2. By our argument, there is no collapse in this loss function. The difference with InfoNCE loss is that the learned feature spreads along the directions of the augmentation $C$, not along the directions of the feature $A_0$.

The case of Barlow Twin (BT) (Zbontar et al., 2021) is similar. While the fourth-order term of BT is much more complicated due to the imbalance created by the $\lambda$ term. The second-order term can be identified easily: $L_{BT} = -2\mathrm{Tr}[W\Sigma W^T] + O(\|W\|^4)$. This also does not collapse. A difference between the SCL loss and InfoNCE is that the learned representation has a spread that aligns with the combination of the feature and the augmentation strength.

## Appendix E. Proofs

### E.1. Proposition 6

Before proving the main results, we first prove a proposition that we will rely on to prove the main results. The following proposition shows that the variance term of the loss takes a specific form when the data is Gaussian.

**Proposition 6** *Let the data and noise be Gaussian. Then, $L = -\mathrm{Tr}[WBW^T] + \mathrm{Tr}[W\Sigma W^T W\Sigma W^T]$.*

*Proof.* The second term in Eq. (4) can be written as

$$\mathrm{Var}[|W(x-\chi)|^2] = \mathbb{E}\left[(\mathrm{Tr}[W(x-\chi)(x-\chi)^T W^T])^2\right] - \mathbb{E}\left[\mathrm{Tr}[W(x-\chi)(x-\chi)^T W^T]\right]^2 \tag{18}$$

$$= [first\ term] - 4\mathrm{Tr}[W(A_0 + C)W^T]^2 \tag{19}$$

$$= [first\ term] - 4\mathrm{Tr}[W\Sigma W^T]^2, \tag{20}$$

where we have used the definition $\Sigma = A_0 + C$. The first term is

$$[first\ term] = \mathbb{E}\left[(\mathrm{Tr}[W(x-\chi)(x-\chi)^T W^T])^2\right] = 4\mathrm{Tr}[W\Sigma W^T]^2 + 8\mathrm{Tr}[W\Sigma W^T W\Sigma W^T]. \tag{21}$$

Combining the above expressions, we see that Eq. (4) can be written as

$$L = -\mathrm{Tr}[WBW^T] + \frac{1}{8}\mathrm{Var}[|W(x-\chi)|^2] \tag{22}$$

$$= -\mathrm{Tr}[WBW^T] + \mathrm{Tr}[W\Sigma W^T W\Sigma W^T]. \tag{23}$$

This finishes the proof. □

### E.2. Proof of Theorem 1

*Proof.* All stationary points have a zero gradient:

$$-2WB + 4W\Sigma W^T W\Sigma = 0. \tag{24}$$

Multiplying by $W^T$ on the left and $B^{-1}$ on the right,

$$W^T W = 2W^T W\Sigma W^T W\Sigma B^{-1} \tag{25}$$

$$(\Longleftrightarrow) \quad \Sigma^{1/2}W^T W\Sigma^{1/2} = 2\Sigma^{1/2}W^T W\Sigma W^T W\Sigma B^{-1}\Sigma^{1/2} \tag{26}$$

Defining $H := \Sigma^{1/2}W^T W\Sigma^{1/2}$, we obtain

$$H = 2H^2\Sigma^{1/2}\Sigma B^{-1}\Sigma^{1/2}, \tag{27}$$

$$(\Longleftrightarrow) \quad H(I - 2H\Sigma^{1/2}B^{-1}\Sigma^{1/2}) = 0. \tag{28}$$

Because both $H$ and $\Sigma^{1/2}\Sigma B^{-1}\Sigma^{1/2}$ are symmetric, one can take the transpose of Eq. (27) to find that $H$ and $\Sigma^{1/2}B^{-1}\Sigma^{1/2}$ commute with each, which implies that $H$ has the same eigenvectors as $\Sigma^{1/2}B^{-1}\Sigma^{1/2}/2$.

Eq. (28) then implies that the eigenvalues of $H$ is either the inverse of that of $\Sigma^{1/2}B^{-1}\Sigma^{1/2}$ or zero. This implies that any stationary point of $H$ can be written in the form

$$H = \frac{1}{2}UM\Lambda U^T, \tag{29}$$

where $U$ is a unitary matrix, $\Lambda$ is diagonal matrix containing the eigenvalues of $\Sigma^{1/2}B^{-1}\Sigma^{1/2}$, and $M$ is an arbitrary (masking) diagonal matrix containing only zero or one such that (1) $M_{ii} = 0$ if $\Lambda_{ii} < 0$ and (2) contain at most $d^*$ nonzero terms. This then implies that the weight matrix $W$ satisfies

$$W^T W = \frac{1}{2}\Sigma^{-1/2}UM\Lambda U^T\Sigma^{-1/2}. \tag{30}$$

Lastly, when $\Sigma$ and $B$ commute, we can compactly write the result as

$$W^T W = \frac{1}{2}\Sigma^{-1}B_M\Sigma^{-1}, \tag{31}$$

where $B_M$ denotes the matrix obtained by masking the eigenvalues of $B$ with $M$. This finishes the proof. □

### E.3. Proof of Proposition 2

*Proof.* For all stationary points, $W^T W$ commutes with $B$ and $\Sigma$, which means that at these stationary points, one can simultaneously diagonalize all the matrices and the loss function (4) can be written as

$$L = -\sum_{i=1}^{d^*} \lambda_i b_i + \lambda_i^2 s_i^2 \tag{32}$$

where $\lambda_i$, $b_i$, $s_i$ are the eigenvalues of $W^T W$, $B$, and $\Sigma$ respectively.

We can thus consider each $i$ separately. When $b_i > 0$, $\lambda_i = 0$ cannot be a local minimum because the local Hessian is $-b_i < 0$. When $b_i \leq 0$, the only stationary point is $\lambda_i = 0$. This sum covers at most $d^*$ summands, and so, at the local minima, $\lambda_i \neq$ if and only if $b_i > 0$, and so the number of non-zero eigenvalues of $W^T W$ is $\min(m, d^*)$. □

# Extended Abstract Track

## E.4. Proof of Proposition 3

*Proof.* The regularization can be written as

$$R = [(\mathbb{E}_x \|Wx\|^2 - c)^2] \tag{33}$$

$$= \text{Tr}[W\Sigma W^T]^2 - 2c\text{Tr}[W\Sigma W^T] + c^2. \tag{34}$$

By Proposition 6, Eq. (5) reads

$$L = -\text{Tr}[WBW^T] + \text{Tr}[W\Sigma W^T W\Sigma W^T] + \kappa(\text{Tr}[W\Sigma W^T]^2 - 2\text{Tr}[W\Sigma W^T] + 1) \tag{35}$$

$$= -\text{Tr}[W(B + 2\kappa c\Sigma)W^T] + \text{Tr}[W\Sigma W^T W\Sigma W^T] + \kappa\rho^2. \tag{36}$$

The derivative of $\rho$ is

$$\frac{d}{dW}\rho = 4\rho W\Sigma. \tag{37}$$

The zero-gradient gradient is thus

$$-2W(B + 2\kappa c\Sigma - 2\kappa\rho\Sigma) + 4W\Sigma W^T W\Sigma = 0. \tag{38}$$

We can define $B' := B + 2\kappa c\Sigma - 2\kappa\rho\Sigma$ to see that this condition is the same as Eq. (24) in the proof of Theorem 1. The rest of the proof thus follows from the arguments. We thus arrive at the theorem statement:

$$W^T W = \frac{1}{2}\Sigma^{-1}B'_M\Sigma^{-1}. \tag{39}$$

We are done. □

## E.5. Proof of Proposition 5

*Proof.* Recalling that $\rho = \text{Tr}[W\Sigma W^T]$, we multiply $\Sigma$ from the right to both sides of the solution in Proposition 3 and take trace:

$$\frac{1}{2}\text{Tr}[\Sigma^{-1}B'_M] = \frac{1}{2}\text{Tr}[\Sigma^{-1}(B_M + 2\kappa(c - \rho)\Sigma_M)] \tag{40}$$

$$= \text{Tr}[W^T W\Sigma] \tag{41}$$

$$= \text{Tr}[W\Sigma W^T] = \rho. \tag{42}$$

The first line further simplifies to

$$\frac{1}{2}\text{Tr}[\Sigma^{-1}B_M] + \kappa(c - \rho)\text{Tr}[\Sigma^{-1}\Sigma_M] = \frac{1}{2}\text{Tr}[\Sigma^{-1}B_M] + \kappa(c - \rho)d_M, \tag{43}$$

where $d_M := \text{Tr}[M]$ is the number of nonzero eigenvalues of $B'_M$.

This gives an equation of $\rho$ that solves to

$$c - \rho = \frac{c - \frac{1}{2}\text{Tr}[\Sigma^{-1}B_M]}{1 + \kappa d_M}. \tag{44}$$

This proves the proposition. □

