# OpenReview forum: "On Rotational Symmetry in the Loss landscape of Self-Supervised Learning"
_NeurIPS.cc/2022/Workshop/NeurReps — NeurReps 2022 Poster_

### Official Review · Reviewer_Hoz6 · 2022-10-10
**Understanding the dimensional collapse of self-supervised learning**

**Confidence:** 3
**Soundness:** 3
**Presentation:** 3
**Contribution:** 3
**Overall Rating:** 7

**Summary:**

The authors derive an analytically tractable theory of SSL landscape and show that it accurately captures an array of collapse phenomena and identifies their causes.

**Questions:**

Based on the theory in the paper, is there an efficient way to prevent dimensional collapse?

**Limitations:**

It would be great to show more examples about landscape of nonlinear models.

**Recommended Decision:**

3: Accept

**Relevance:**

3: Solid fit

**Strengths And Weaknesses:**

The dimensional collapse is a very important problem in SSL. The authors develop the landscape theory to study the failure mode of dimensional collapse. They provide an interesting insight that collapses of representations is strongly dependent on the stability of the last layer at the origin and happens when a broken symmetry is restored. Lots of numerical studies are provided to support the theory.


**Submission Track:**

Extended Abstract (4 Page)

---

### Official Review · Reviewer_M6P6 · 2022-10-12
**Review of "What shapes the loss landscape of self-supervised learning?"**

**Confidence:** 2
**Soundness:** 4
**Presentation:** 3
**Contribution:** 3
**Overall Rating:** 7

**Summary:**

This paper theoretically analyzes the loss landscape of self-supervised learning, both in general and for specific common loss functions, towards the understanding of various types of collapse in linear and non-linear models.

**Questions:**

Proofreading:
“...collapses of representations is strongly dependent…” -> ”...collapses of representations are strongly dependent…”

“Different from InfoNCE, one terms in the…” -> “Different from InfoNCE, one term in the…”

“...and (2) contain at most $d^*$ nonzero terms.” -> “the number of nonzero terms is at most $d^*$.”

“...and let $\Lambda_i$ be the eigevalues…” ->  “...and let $\Lambda_i$ be the eigenvalues…”

“...and Barlow Twins, which discuss in Appendix D.4 “ -> “...and Barlow Twins (Zbontar et al., 2021), which are each discussed in Appendix D.4.“

“...for any rotation matrix $R$, $L(x)=L(Rf(x)$.” -> “...for any rotation matrix $R$, $L(x)=L(Rf(x))$.”

Some citations are inconsistent, such as citing works from NeurIPS sometimes as proceedings and sometimes as articles, as well as inconsistent ways of writing the venue.



**Limitations:**

The authors briefly address the limitations of using linear vs. non-linear models in theory, both in their work and in prior works, although these discussions are mainly in the supplementary work.


**Recommended Decision:**

3: Accept

**Relevance:**

2: Limited relevance

**Strengths And Weaknesses:**


The submission seems novel and technically sound to my knowledge, although I am not an expert in this subject matter. The experimental results provided serve as further evidence for the theoretical results and claims.

Regarding clarity, the structure of the paper makes it feel incomplete. What is currently the first main paragraph is similar to what the abstract should be. Section 1, which constitutes the remainder of the main paper, should be split into sections rather than subsections. It also ends rather abruptly: a concluding paragraph could help tie the sections together by summarizing the final conclusions.

The treatment of $\kappa$ is a bit unclear: in Section 1.2, you hint twice that you are looking at the limit of $\kappa \to \infty$, but then finally refer to an appendix, with the work in between these “hints” not yet considering this limit.  The second half of the second paragraph in Section 1.2 could be removed, as it describes the work in the fourth paragraph in almost as many words, without clear wording that it is meant to just introduce the following ideas.

The results seem important for a broad audience, although the relevance to this workshop is only minor.



**Submission Track:**

Extended Abstract (4 Page)

---

### Official Review · Reviewer_E373 · 2022-10-14
**Insightful analytical work analyzing common modern self-supervised learning loss functions using linear theory**

**Confidence:** 3
**Soundness:** 4
**Presentation:** 3
**Contribution:** 4
**Overall Rating:** 8

**Summary:**

This work follows a popular formula for gaining insight into deep learning: analyzing in full analytical detail the dynamics of linear networks. The linear assumption allows a full analytical characterization of the setup and can help elucidate under what conditions trivial solutions are found. In self-supervised learning, this analysis reveals the causes of different types of representational collapse, which is very valuable.

**Questions:**

It would be helpful to have a table with mathematical quantities and what they exactly mean.

I am also a big believer of simple geometric diagrams to improve clarity, intuition and reach and I hope the authors do this for their final submission (and poster) - It would also be useful to have schematics of the failure modes and indicate which term in the simplified losses are causing the different collapses to make it more intuitive without having to go through all the equations (for the less mathematical folks). This schematic could be repeated and modified to show what other changes discussed like the addition of bias and the addition of normalization do.

**Limitations:**

The authors have discussed limitations of their work.

**Recommended Decision:**

3: Accept

**Relevance:**

4: Highly relevant

**Strengths And Weaknesses:**

Strengths:
The analysis is simple, upon fully understanding the work (which took some time, see weaknesses below), the math became intuitively clear to me and the authors have demonstrated well that their simple linear analysis works for actual networks such as ResNet trained Cifar-10. The authors also comment on how other factors such as normalization, non-linearity and the addition of a bias affect their results. In the appendix, the authors have thoroughly and convincingly explored how robust their results are numerically and have provided (almost) full proofs with missing steps of their main results.

Weaknesses:
- The paper is hard to read. Many terms in mathematical equations are stated without being defined clearly. I had to struggle a lot and read through the appendix in detail to follow. For example, $\chi$ has not been defined and is used throughout the paper, started at equation 1.

- The analysis is presented and worked out in detail for contrastive InfoNCE and its variants, and but there is not any information (beyond a single paragraph in appendix D.4 for Barlow twins which I did not follow) showing how it generalizes to other types of non-contrastive loss function. I would like to see at least a short overview of how it generalizes to these non-contrastive terms as well.

**Submission Track:**

Extended Abstract (4 Page)

---

### Decision · Program_Chairs · 2022-10-21

Accept (Poster)